# A Frailty-Adjusted Stratification Score to Predict Surgical Risk, Post-Operative, Long-Term Functional Outcome, and Quality of Life after Surgery in Intracranial Meningiomas

**DOI:** 10.3390/cancers14133065

**Published:** 2022-06-22

**Authors:** Leonardo Tariciotti, Giorgio Fiore, Sara Carapella, Luigi Gianmaria Remore, Luigi Schisano, Stefano Borsa, Mauro Pluderi, Marco Canevelli, Giovanni Marfia, Manuela Caroli, Marco Locatelli, Giulio Bertani

**Affiliations:** 1Unit of Neurosurgery, Foundation IRCCS Cà Granda Ospedale Maggiore Policlinico, 20122 Milan, Italy; giorgio.fiore@unimi.it (G.F.); sara.carapella@gmail.com (S.C.); luigigianmaria.remore@gmail.com (L.G.R.); luigi.schisano@policlinico.mi.it (L.S.); stefano.borsa@policlinico.mi.it (S.B.); mauro.pluderi@policlinico.mi.it (M.P.); giovanni.marfia@unimi.it (G.M.); manuela.caroli@policlinico.mi.it (M.C.); marco.locatelli@unimi.it (M.L.); giulio.bertani@policlinico.mi.it (G.B.); 2Department of Oncology and Hemato-Oncology, University of Milan, 20122 Milan, Italy; 3Department of Human Neuroscience, Sapienza University, 00185 Rome, Italy; marco.canevelli@uniroma1.it; 4National Center for Disease Prevention and Health Promotion, National Institute of Health, 00161 Rome, Italy; 5Laboratory of Experimental Neurosurgery, Unit of Neurosurgery, Foundation IRCCS Cà Granda Ospedale Maggiore Policlinico, 20122 Milan, Italy; 6Department of Pathophysiology and Transplantation, University of Milan, 20122 Milan, Italy; 7“Aldo Ravelli” Research Center for Neurotechnology and Experimental Brain Therapeutics, University of Milan, 20122 Milan, Italy

**Keywords:** meningioma, prognostic score, skull base surgery, frailty index, quality of life, functional assessment

## Abstract

**Simple Summary:**

Meningiomas are the most common primary brain tumour and their prevalence increases in the ageing population. Among researchers, predicting surgical outcomes, complications, and quality of life (QoL) after surgery still represents a major subject of debate. The aim of the study hereby presented was to review well known and potential new prognostic factors affecting the early and long-term functional outcomes and quality of life of patients treated for intracranial meningiomas. Our findings might help define tailored surgical and perioperative protocols to maximise the standard of care, relying on a patient-specific multi-domain surgical, biometric, and clinical assessment to be conducted during the pre-operative medical interview. This approach might be beneficial for reducing complications occurrence, predicting surgical and functional outcomes, counselling patients and caregivers on surgical indications, reducing legal issues, and providing a valuable tool to healthcare providers for resources allocation.

**Abstract:**

Object: To investigate those parameters affecting early and follow-up functional outcomes in patients undergoing resection of meningiomas and to design a dedicated predictive score, the Milan Bio(metric)-Surgical Score (MBSS) is hereby presented. Methods: Patients undergoing transcranial surgery for intracranial meningiomas were included. The most significant parameters in the regression analyses were implemented in a patient stratification score and were validated by testing its classification consistency with a clinical–radiological grading scale (CRGS), Milan complexity scale (MCS), and Charlson Comorbidity Index (CCI) scores. Results: The ASA score, Frailty index, skull base and posterior cranial fossa locations, a diameter of >25 mm, and the absence of a brain–tumour interface were predictive of early post-operative deterioration and were collected in MBSS Part A (AUC: 0.965; 95%C.I. 0.890–1.022), while the frailty index, posterior cranial fossa location, a diameter of >25 mm, a edema/tumour volume index of >2, dural sinus invasion, DWI hyperintensity, and the absence of a brain–tumour interface were predictive of a long-term unfavourable outcome and were collected in MBSS Part B (AUC: 0.877; 95%C.I. 0.811–0.942). The score was consistent with CRGS, MCS, and CCI. Conclusion: Patients’ multi-domain evaluation and the implementation of frailty indexes might help predict the perioperative complexity of cases; the functional, clinical, and neurological early outcomes; survival; and overall QoL after surgery.

## 1. Introduction

Meningiomas are the most common primary intracranial tumours and their incidence increases in patients older than 65 years [1]. The incidence of intracranial meningiomas in the general population is 1:12,500, but it increases exponentially with ageing, yielding a 1:2000 incidence in patients older than 80 years [2].

The growing geriatric proportion in general populations of high-income countries will soon represent a new century challenge for surgeons: it is estimated that by 2040, the population aged over 65 years will increase by two-fold [3]. The aforementioned trend suggests that a new decision-making process should be attempted when these patients’ eligibility for cranial surgery is evaluated.

A significant proportion of cases (90%) are benign (WHO grade I) and potentially curable once gross total resection is performed [4]; however, 5–7% of cases represent atypical (WHO grade II) and 1–3% anaplastic meningiomas (WHO grade III) [1,3]. The latter histology and unfavourable anatomical location results in a significant risk of subtotal resection or recurrence, and adjuvant treatments have a relevant impact on the long-term outcomes in these circumstances [5].

As an effect of improved technologies, modern surgical techniques, and perioperative optimised management, eligibility for the microsurgical resection of intracranial meningiomas has become feasible in patients with advanced age, affected by several comorbidities or tumours in previously un-attempted anatomical locations. The higher the odds of complications, the more reduced the patient “biological compliance and reserve” and the higher the costs for healthcare providers make the prediction of surgical outcomes essential in these cases. Frailty indices—surrogates of biological reserve against stressors—have been proposed as predictors of morbidity and mortality after surgery for brain tumours, intracerebral haemorrhages, and spine surgeries [6,7,8,9,10,11,12,13], especially on national or insurance-related registries. These studies rely on age, functional status, physician subjective impression, or a combination of the previous parameters to define whether patients bore a frail profile or not, rather than testing a validated frailty index in several cases. The latter was received as hard to compute in clinical practice, as the investigated variables were frequently missing in large retrospective datasets [14], preventing their routine use in surgical wards.

The quality of life (QoL) quantitative assessment was recently used to describe long-term outcomes in patients undergoing surgery for brain tumours; however, it yielded inconsistent findings in patients affected by meningiomas. Previous series reported improved QoL after surgical resection of meningiomas [15,16,17], while a recent prospective longitudinal study highlighted long-term impairment in quality of life after tumour resection [18]. Despite recent attempts, the use of patient-reported outcome measures, QoL, and daily independence (a surrogate of better post-operative overall cognition and reduced neurological deficits) have been poorly investigated in patients undergoing surgery for intracranial meningioma [19,20,21,22].

We designed a specific risk score for patients undergoing intracranial meningiomas resection to predict the overall complexity of surgery, improve patient care, improve resource allocation, select senior skilled surgeons, or refer the case to a neurological institute with a higher caseload when advisable.

## 2. Materials and Methods

### 2.1. Study Type and Participants

The study was a monocentric retrospective observational study conducted at the “Unit of Neurosurgery, Foundation IRCCS Cà Granda Ospedale Maggiore Policlinico” in Milan, Italy, from December 2016 to December 2020. Ethical approval was waived as per the institutional policy on retrospectively designed studies. The investigation was conducted in accordance with the principles of the Declaration of Helsinki.

All patients admitted for elective trans-cranial surgery diagnosed with intracranial meningiomas in the interval between December 2016 and December 2020 were included. The exclusion criteria were as follows: (a) patients with lesions other than meningiomas in the brain or spine, (b) recurrent intracranial meningioma, (c) patients diagnosed with neurofibromatosis type 1 or 2, (d) <95% data availability during electronic registries review and follow-up interview, and (e) loss at follow-up.

### 2.2. Variables of Interest and Outcomes

Pre- and post-operative demographic, clinical, and radiological characteristics and relevant complication occurrence were extracted from hospital electronic medical records. Anatomical location was reported according to previous classifications [23]. Radiological features were evaluated on pre-operative MRI scans deposited in the institutional picture archiving and communication system (PACS) database. The grading and biological behaviour description were conducted according to the 2016 World Health Organization (WHO) classification of Tumors of the Central Nervous System [24] by experienced neuropathologists.

#### 2.2.1. Biometric/Functional Data Extraction

The American Society of Anesthesia physical status classification system (ASA) was extracted from electronic medical records, as attributed by a senior neuro-anaesthesiologist. The Karnofsky performance scale (KPS) [25,26] was measured upon admission, discharge, and follow-up interview: the cut-off value considered in the current investigation was dichotomised as >70 and ≤70, as this value marks independent vs. dependent functional status [25,26].

A 34-items Frailty Index (FI) score was computed according to previous literature [27,28]. All of the parameters were recorded as a binary variable (e.g., presence vs. absence of the deficit); the sum of all of the deficits was calculated and a frailty index score ranging from 0 to 1 point was computed (n° positive items/34). Patients with scores <0.10 were considered “fit” (not frail), scores within 0.11–0.20 “semi-fit”, and scores >0.20 were considered “frail”. The cut-offs applied in the current investigation were previously investigated [27,29]. The 34-item frailty index defined and implemented in the current study is shown in Table 1.

#### 2.2.2. Radiological Data Extraction

Anatomical location, pre- and post-operative tumour volume, peritumoral edema volume (PTE), and the extent of resection (categorised as gross total resection (GTR) or subtotal resection (STR)) were defined by evaluating the pre-operative and three-month post-operative brain MRIs. All of the tumours were processed employing manual segmentation and volumetric assessment by a senior neurosurgeon (G.B) using Horos^®^. Horos is a free and open-source code software (FOSS) program that is distributed free of charge under the LGPL license at Horosproject.org and is sponsored by Nimble Co LLC d/b/a Purview in Annapolis, MD, USA. Severe peritumoral edema was stated as edema index (EI) = edema volume/tumour volume ratio >2.0 [30].

Maximal tumour diameter was stated as the maximal diameter measured on all three axes on a T1 MRI scan after gadolinium enhancement (Gadovist 0.1 mL/kg; Prohance 0.2 mL/kg) administration. The presence of necrosis, hyperostosis, heterogeneous contrast enhancement, dural sinus invasion (according to Sindou et al. [31]), a tumour–brain cleft in T2 weighted scans, and DWI hyperintensity compared with parenchymal grey matter (b-values of 0–1000 s/mm^2^) were documented by senior neuroradiologists at our institution. An irregular tumour shape was defined per multilobulated tumours (>2 lobules) [32].

#### 2.2.3. Surgical Management

Microsurgical resection was performed under general anaesthesia and with intraoperative neuronavigation (Brainlab^®^) assistance. Intraoperative fluorescence for vessel enhancement or brain/tumour interface visualisation, intraoperative neurophysiological mapping, and monitoring were used according to the surgeon’s preference.

#### 2.2.4. Post-Operative and Follow-Up Data Extraction

##### Early Post-Operative and Long-Term Parameters of Interest

The post-operative variables were: tumour grade, intensive care unit (ICU) in-stay ≥ 24 h, post-operative haemorrhage, infections occurrence, and discharge to a rehabilitation facility. All of the complications were rated according to the Clavien Dindo classification [33]. At the long-term follow-up interview, KPS and the Functional Assessment of Cancer Therapy brain subscale (FACT-Br) were used to measure physical, social, family, emotional, and functional well-being and the overall quality of life during follow-up interview [34]. Overall mortality at 30 days and 1, 3, and 5 years were also measured.

##### Primary Post-Operative and Follow-Up Outcomes

We designed two specific parameters to dichotomise the early post-operative and the long-term functional outcomes for further analyses:(1)The primary post-operative outcome (“early post-operative functional deterioration”) was designed to address patient dependence status and was computed as a drop in post-operative KPS of at least 20 points at discharge compared with the pre-operative assessment. The selected cut-off represented the occurrence of any general or neurological complication affecting the overall functional performance of patients undergoing surgery well [21,35].(2)The long-term follow-up primary outcome (“unfavourable long-term functional autonomy and quality of life”) was designed to address patient dependence and QoL and was defined as a decrease of ≥ 20 points in KPS at the follow-up interview compared with the pre-operative one plus an overall quality of life (QOL) under the 75th percentile of the examined population.

### 2.3. Statistical Analysis

Descriptive statistics, frequencies, and percentages were used to report demographic, clinical, and radiological variables. A Shapiro–Wilk normality test was used to assess normality across the selected variables. When appropriate, continuous variables were reported as mean + standard deviation (SD) or median and interquartile range (IQR). Ordinal and categorical variables were reported as absolute numbers and percentages (N; N%).

First, a logistic univariate regression analysis was carried out for all variables under investigation. Significant predictors were then entered into a logistic multivariable regression model. The odd ratio (OR), Nagelkerke R^2^, and Akaike Information Criterion (AIC) were used to evaluate the goodness of fit.

The independent predictors were employed to define a prognostic score. Each patient was finally rated according to the predictors enlisted. A Mann–Whitney test was performed to compare the mean score between classes of dichotomised outcomes.

A receiver operating characteristic analysis (AUC-ROC) was performed to examine the areas under the curve (AUCs), standard error, and 95% confidence interval (95% CI) of the prognostic score for classifying post-operative and long-term outcomes, respectively. A secondary AUC-ROC analysis was conducted to compare the classification performance of the proposed score, clinical–radiological grading system (CRGS), Milan complexity scale (MCS), and Charlson comorbidity index (CCI) scores. The AUC for FI was computed for comparison. The cumulative risk of impairment after surgery for each point of increase in MBSS was reported according to the predictors’ relative risks.

Exploratory analysis was performed using SPSS (version 27.0, IBM, Armonk, NY, USA) and Python programming language (version 3.5; libraries: Sklearn, Matplotlib, Seaborn, Pandas, and Scipy). Regression and ROC analyses were performed using SPSS.

## 3. Results

### 3.1. Population Description

A total of 165 patients were included. The mean patient age was 62 ± 13 and 116 were female (70.3%). The median KPS was 90 (IQR 80–90). Post-operative functional deterioration was noted in 22 patients (13.3%): 9 (5.5%) patients suffered from a post-operative complication requiring re-intervention (Clavien–Dindo classification grade ≥3). The 30-day mortality was 1.8%. 

At follow-up, the median KPS was 90 (IQR 80–90). Median QoL assessed through the FACT-Br subscale was 169.57/300.00 (IQR 157.93–183.06). Twenty-eight (17%) patients experienced long-term unfavourable functional outcomes and an assessment of QoL below the 75th percentile (QoL < 183 points) at the follow-up interview.

Mortality at 6 months and 1, 3, and 5 years was 1.82%, 3%, 4.8%, and 5.5% of the overall population, respectively. All patients were clinically and radiologically followed-up (median follow-up interval: 33 months; IQR 18–48) and outpatient evaluations were carried out.

Additional demographic details and complications are summarised in Table 2.

### 3.2. Regression Analysis of Early/Long-Term Post-Operative Functional Outcome and Score Design

Univariate and multivariable regression analyses fitting the clinical, biometric, and radiological features were performed to test any potential association with early post-operative functional deterioration (KPS decrease after surgery ≥ 20) and long-term unfavourable functional outcome (KPS ≥ 20 decrease at follow-up compared with baseline and overall assessed QOL under the 75th percentile). The results are reported in Table 3. A prognostic score for early post-operative (Part A) and long-term functional outcomes (Part B) was developed by listing and rating significant predictors according to ORs at the multivariable regression analysis, maintaining a constant importance ratio among predictors (Table 4).

#### 3.2.1. MBBS Part A

A grading scale ranging from 0 to 15 was built to assess the risk of post-operative functional deterioration. The parameters included were as follows: ASA score, frailty index, skull base and infratentorial location, diameter of >25 mm, and the absence of a tumour–brain cleft. A higher MBBS Part A score represented a more complex surgical case prone to increased risk of post-operative functional deterioration. The MBBS Part A is shown in Table 4.

Patients experiencing a worsening of their functional status after surgery showed a significantly higher MBBS Part A score compared with those improving or remaining stable (12 [IQR 2–19] vs. 4 [IQR 2–7]; *p* < 0.001; Figure 1A). Patients requiring longer (>12 h) post-operative ICU monitoring (8 [IQR 3–12] vs. 4 [IQR 2–7]; *p* < 0.001); patients with post-operative intraparenchymal haemorrhage (6 [IQR 3–7] vs. 4 [IQR 2–7]; *p* ≤ 0.009); and those experiencing any severe post-operative complication requiring surgical intervention, ICU re-admission, or deceased after surgery (10 [IQR 7–12] vs. 4 [IQR 2–7]; *p* < 0.001) showed a significantly higher MBBS score Part A. After analysing the mortality rates in the population investigated, higher MBBS-A scores were reported in patients deceased within 30 days after surgery (12 [IQR 12–12] vs. 4 IQR [2–7]; *p* = 0.003).

#### 3.2.2. MBBS Part B

To assess the risk of long-term unfavourable functional outcome, a grading scale ranging from 0 to 10 was designed. The parameters included were as follows: frailty index, infratentorial location, diameter of >25 mm, severe peritumoral edema, sinus invasion, intra-tumoral DWI hyperintensity, and the absence of a tumour–brain cleft. A higher MBBS Part B score indicates a higher risk of long-term suboptimal functional outcome and reduced quality of life. MBBS Part B is shown in Table 4.

Patients reporting an unfavourable long-term functional outcome and reduced QOL at follow-up evaluation/interview scored significantly higher in MBBS Part B compared with those reporting a favourable outcome (5 [IQR 4–6] vs. 2 [IQR 1–3]; *p* < 0.001; Figure 1B). Analysing mortality rates in the population investigated, higher MBBS-B scores were reported in patients deceased within 6 months and (6 [IQR 5–6] vs. 3 [2–4]; *p* < 0.001), 1 (6 [IQR 5–6] vs. 3 [2–4]; *p* = 0.001), 3 ((5 [IQR 3–6] vs. 3 [2–4]; *p* = 0.011), and 5 years (4 [IQR 4–6] vs. 3 [2–4]; *p* = 0.013) after surgery.

### 3.3. Internal Retrospective and External Comparative Validation

Receiver operating characteristic (ROC) analyses were computed for MBSS-A and MBSS-B to assess the classification performance of the scores. The scores were tested on the overall population and age-based subgroups. The results are reported in Table 4 and the cumulate risk per single point increase is described in Appendix A.

Finally, the clinical–radiological grading system (CRGS), Milan Complexity Scale (MSC), and Charlson comorbidity index (CCI) were calculated on the population under exam as prognostic reference scores. The median CRGS was 16 [IQR 14–17], the mean MSC was 1.33 ± 0.160, and the median CCI was 3 [IQR 1–4].

An AUC-ROC analysis was computed to compare the overall performance of MBSS-A, MBSS-B and CRGS, MCS, and CCI. The results are summarised in Table 5 and Figure 2.

A checklist explaining how to reproduce MBSS Part A and Part B and a definition of each variable relevant to the definition of the score is shown in Appendix A.

## 4. Discussion

Meningiomas represent the most frequent primary intracranial tumours and among all, because of their slow-growing attitude, low biological aggressiveness, and surgical curability, they are generally prone to a benign course. The population affected is frequently represented by elderly patients and the possibility of predicting post-operative and long-term functional outcomes is still a fostered research topic.

FIs have been widely adopted in medical and surgical practice in recent years as quantifiers of cumulative personal morbidity, but still represent a novel concept in daily neuro-oncological routine. Despite 11-, 8-, and 5-item modified FI scores having been proposed, whether they reliably measure frailty or not is still under debate in the scientific community [28,29,36]. Reducing the operators of the score to provide a more clinically-appealing tool is to be pursued, unless the consistency of the instrument is undermined. Indeed, the publication of “FI sub-scores” was received as a consequence of the exclusion of specific pre-operative variables—previously computed in the 11-items Mfi—from the American College of Surgeons National Surgical Quality Improvement Program (ACS NSQIP) database after 2012 [37]. If so, the selection of accurate predictors might have been obligated by requirements in data acquisition instead of clinical methodological insights [38].

The functional outcome was historically measured as the variation of KPS score after surgery: although it depicts patient independence status, it is belittling when QoL is questioned. The latter represents a multidimensional derivative construct comprehending both physical and psychosocial factors that KPS cannot address [39]. FACT-Br represents a validated tool for patients with primary CNS tumours and is widely accepted as a more specific measure of long-term satisfactory functional outcome and quality of life [34].

Among the predictors included in our score, the ASA score was statistically associated with post-operative KPS reduction, but not with long-term functional results. As it represents a valuable tool to assess patient pre-anaesthesia comorbidities, an association between ASA score and long-term functional outcome was conceptually weak and our findings confirmed this empiric assumption.

FI showed a strong association with both post-operative and long-term functional outcomes. Interestingly, a pre-operative moderate FI (FI = 0.11–0.20) showed a higher impact on long-term than early functional outcomes: this could be justified by an additional loss in physiological reserve (i.e., an unmeasured increase in FI) during follow-up, affecting independence and QoL.

The association of tumour anatomical and biological characteristics to post-operative unfavourable results reside in additional technical difficulties regarding microsurgical devascularisation and dissection of the meningioma from the dural base and arachnoid, which puts patients at risk of prolonged in-ward stay, severe post-operative complications, brain infarction, and overall increased re-operation rate. An early unfavourable surgical outcome increases the risk of functional and cognitive patient impairment, which has a lasting effect on long-term independence and QoL.

In recent years, several proposals of pre-operative scores and models able to predict surgical outcomes after brain tumour surgeries have been published: the clinical-radiological grading system (CRGS) [40] was designed and validated at our institution more than 15 years ago by Caroli et al., and the Milan complexity scale (MCS) [35], the SKALE system [41], the Charlson comorbidity index score (CCI) [42], and the Geriatric scoring system (GSS) [43] have been described and implemented on patients with meningiomas.

Despite wide application and validation, these scores exhibit a variable number of limitations: CCI is a geriatric assessing tool but might perform poorly on neurosurgical cohorts, as suggested by our comparative validation analysis; CRGS, GSS, and SKALE systems are affected by a qualitative categorisation of parameters that might affect the overall generalizability of results across different populations. MCS is a highly effective resource with large-scale implications in indexing case complexity for perioperative management or case referral. Still, its design limits application when the prediction of post-operative functional status and quality of life is demanded. Moreover, it was designed on a heterogeneous cohort of CNS tumours with different biological courses and surgical technical peculiarities.

We designed a predictive score collecting the points of strength of the previous scores and implemented a patient-centred biometric assessment of biological reserve through a dedicated frailty index score.

This process resulted in a multi-domain biological, functional, and surgical integrating score, the Milan Biometric-Surgical Score for meningiomas (MBSS-Men Part A and B).

Finally, the comparative classification analysis of MBSS, MCS, CRGS, and CCI provided insight into model consistency, and—interestingly—FI alone reported a high performance when compared with other scores: its implementation in MBSS might explain the results reported in Table 5 and visualised in Figure 1.

Regardless of the impact of our score on clinical practise, we believe that the role of FI in affecting post-surgical outcomes should be further investigated.

### 4.1. Points of Strength

MBSS was the first score ever designed to implement a frailty index with a weighted importance as high as surgical and anatomical parameters have on a homogeneous cohort.

Predicting the risk of an unfavourable surgical outcome for patients diagnosed with intracranial meningioma through a reliable but easy-to-use pre-operative tool might represent a step forward in clinical practice for a larger audience of neurosurgeons. However, further prospective investigations are required to validate these preliminary results.

### 4.2. Limitations

Our study has several limitations: first, MBSS was built post hoc using a monocentric retrospective registry and was never applied on hold-out testing samples, and for these reasons needs to be prospectively validated; also, the population sample was limited and our findings might not be generalisable to larger cohorts. Furthermore, the reliability of the score on cohorts of patients treated at different institutions should be verified. The use of a full-scale (34 items) frailty index score might discourage others from implementing it: in our experience, the FI checklist can be filled out during the admission interview with no dedicated additional time by means of a smartphone or tablet within 5 min. Finally, our score was defined and tested on a population of trans-cranially resected meningiomas: the results we report might not fit skull base meningiomas treated through endoscopic-assisted skull base corridors. Multicentric validation is recommended.

## 5. Conclusions

Multifactorial consideration of patients suffering from intracranial meningiomas, with particular attention to neurological and biological assessments, will help predict the perioperative complexity of cases; the functional, clinical, early, and late neurological outcomes; survival; and QoL after surgery.

## Figures and Tables

**Figure 1 cancers-14-03065-f001:**
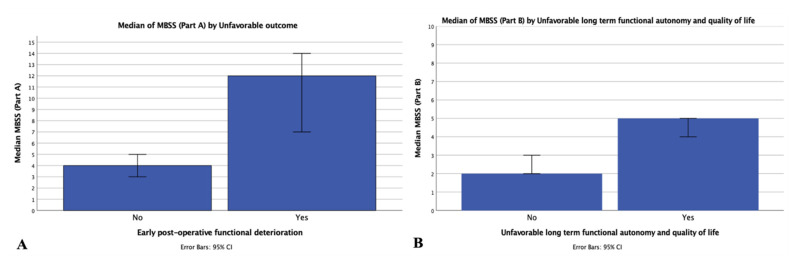
MBSS Part A and B by outcome subgroups. Histograms describe the distribution of MBSS subscores among those patients experiencing or not experiencing an early post-operative or long-term functional deterioration and reduced quality of life. (**A**) The histogram on the left represents the median MBSS Part A scores among those experiencing a decrease of at least 20 points in post-operative KPS or not compared with the pre-operative assessment. 95% CI errors bars are visualised. (**B**) The histogram on the right represents the median MBSS Part B scores among those experiencing a decrease of at least 20 points in follow-up KPS plus an overall quality of life measurement (FACT-Br overall score) under the 75th percentile of the population under exam or not compared with the pre-operative assessment. 95% CI error bars are visualised. MBSS: Milan Biometric-Surgical score; KPS: Karnofsky performance status; CI: confidence interval; FACT-Br: Functional Assessment of Cancer Therapy brain subscale.

**Figure 2 cancers-14-03065-f002:**
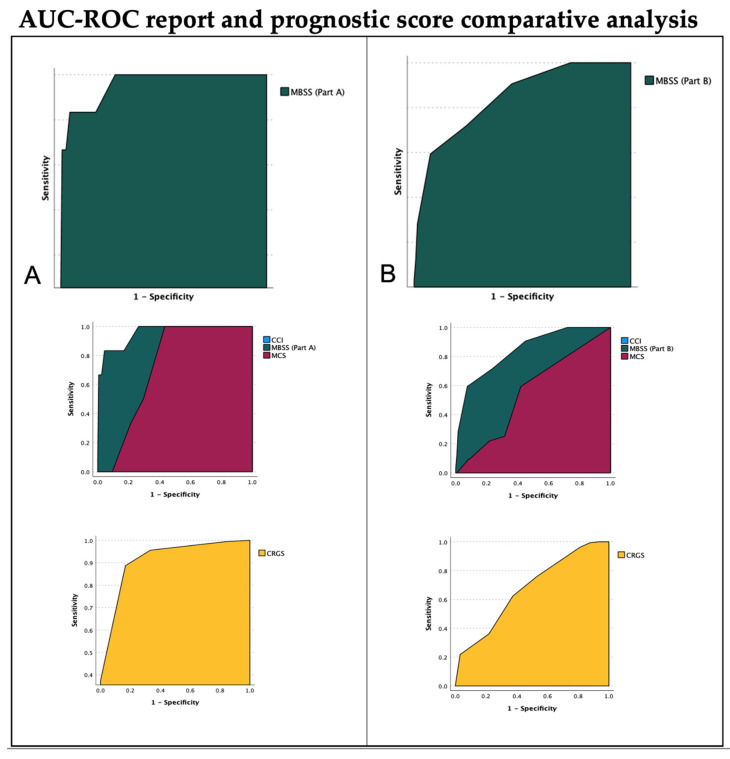
Score AUC-ROC analysis. The area under the curve at the receiver operating characteristic analysis (AUC-ROC) of the MBSS, its sub-scores, and previously validated prognostic scores are shown. (**A**) The AUC of MBSS Part A is reported in the higher part of the figure. In the middle, the AUCs of MCS, CCI (superimposed to MCS and not graphically visualised), and MBSS-Part A are shown for comparison. In the lower image, the AUC-ROC of CRGS is visualised. The AUC-ROC testing variable was a post-operative functional deterioration as defined in the manuscript. MBSS Part A provided the highest AUC in the current analysis. (**B**) The AUC of MBSS Part B is reported in the higher part of the figure. In the middle, the AUCs of MCS, CCI (superimposed to MCS and not graphically visualised), and MBSS-Part B are shown for comparison. In the lower image, AUC-ROC of CRGS is visualised. The AUC-ROC testing variable was a reduction of KPS at follow-up of at least 20 points compared with baseline, plus a quality of life assessment under the 75th percentile. MBSS Part B provided the highest AUC in the current analysis. MBSS: Milan biometric surgical score; CCI: Charlson comorbidity index; MCS: Milan complexity scale; CRGS: clinical–radiological grading system.

**Table 1 cancers-14-03065-t001:** Frailty Index.

**34-Items Frailty-Index for Pre-Operative Assessment in Craniotomy Surgery**							
**Items**	**Code**	**YES**	**NO**	**Items**	**Code**	**YES**	**NO**
1—Smoking status	**Smoking**			18—Thyroid disease	**Thyroid**		
2—Balance disorders	**Balance**			19—Cancer	**Cancer**		
3—Osteoporosis	**Osteop**			20—Chirrhosis	**Liver**		
4—Arthritis/Deformant arthrosis	**Bone**			21—Urinary or bowel incontinence	**Incontinence**		
5—Hypertension (>140/90 mmHg)	**HTN**			22—Stayed in bed > half of the day due to health (last month)	**Bed**		
6—Ischemic heart disease, CAD, PAD	**Ischemia**			23—Parkinsonism	**Park**		
7—Chronic heart failure (CHF)	**Heart**			24—Focal neurological signs	**Neuro**		
8—Arrhythmia	**I**			25—Hearing impairment	**Hearing**		
9—COPD or other respiratory disorders	**Lung**			26—Mobility disability (200 m walking test)	**Mobility**		
10—History of previous blood clot (DVT, PE, TIA or Stroke)	**Clot**			27—Depression (feeling downhearted/depressed most of the time)	**Depression**		
11—Bleeding disorders (thrombocytopenias, NOAC, VKA, other haematological conditions)	**Bleed**			28—Anxiety	**Anxiety**		
12—Dislipidaemia	**Lipids**			29—Sleep disorders (difficulty sleeping > 6 h or takes sleep pilhs)	**Sleep**		
13—Obesity (BMI > 30) or underweight (BMI < 18.5)	**Obesity**			30—Haemoglobin (<13.5 g/dL in males, <12.0 g/dL in females)	**HB**		
14—Gastric disorder	**Gastric**			31—HCT < 26%	**HCT**		
15—Intestinal disorder	**Bowel**			32—Creatinine (<0.6 mg/dL)	**Creatinine**		
16—Diabetes	**DM**			33—Albumin (<3.5 g/dL)	**Albumin**		
17—Chronic kidney disease	**Renal**			34—White blood cells (<4 × 10^3^/mm^3^)	**WBC**		
**Total**					**/34**		

The table represents the frailty index (FI) chart as designed and implemented at our institution. Thirty-four items were selected among historical clinical information, comorbidities, and laboratory test results related with the aging phenomenon. Trained personnel filled the FI chart for each patient during medical interview at admission. The total number of positive items were then divided by 34 to obtain the overall FI score. CAD: carotid artery disease; PAD: peripheral artery disease; DVT: deep venous thrombosis; PE: pulmonary embolism; TIA: transient ischemic attack; NOAC: novel oral anticoagulation; VKA: vitamin K antagonist.

**Table 2 cancers-14-03065-t002:** Demographic characteristics and complications (N = 165 patients).

**Variables**	**Median (IQR)**	**Count (N)**	**N%**	**Variables**		**Median (IQR)**	**Count (N)**	**N%**
**Age**				**Clavien-Dindo Classification Grade**	**No complication**		82	49.70%
**Overall population Age**	63 (52–72)				**Grade 1**		13	7.88%
**18–64**		87	52.41%		**Grade 2**		61	36.97%
**65–79**		60	36.14%		**Grade 3**		5	3.03%
**>80**		18	10.84%		**Grade 4**		1	0.61%
**Gender**					**Grade 5**		3	1.82%
**Female**		116	70.30%	**Length of stay (LOS)**		11 (8–16)		
**Male**		49	29.70%	**ICU discharge > 24 h**			14	8.48%
**Anatomical location**				**Post-operative KPS**		90 (80–90)		
**Convexity**		61/165	37.58%	**Unfavorable out–ome—Post-operative**			22	13.33%
**Parasagittal**		6/165	3.64%	**Operation time (min)**		216 (155–310)		
**Falx**		19/165	11.52%	**ICH**			36	21.82%
**Tentorium**		3/165	1.82%	**Seizure**			7	4.24%
**Cerebellar convexity**		2/165	1.21%	**Infections**			8	4.85%
**CPA**		9/165	5.45%	**Pulmonary embolism**			43	26.06%
**Sphenoid wing**		9/165	5.45%	**Post-operative tumor volume (mL)**		0.41 (2,12)		
**Tuberculum/Dorsum sellae/Planum/Clinoid**		10/165	6.06%	**Gross total resection (GTR)**			128	81.01%
**Middle Fossa**		16/165	9.70%	**KPS at follow-up**		90 (80–90)		
**Olfactory Groove**		20/165	12.12%	**Unfavourable outcome at FU**			28	16.97%
**Clival/Petroclival**		8/165	4.85%	**Mortality:**				
**Foramen Magnum**		1/165	0.61%	**30-day mortality**			3	1.82%
**Intraventricular**		1/165	0.61%	**6-month mortality**			5	3.03%
**WHO grade**				**1-year mortality**			5	3.03%
**I**		126	79.25%	**3-year mortality**			8	4.85%
**II**		31	19.50%	**5-year mortality**			9	5.45%
**III**		2	1.26%	**Functional and patient-reported assessment (FACT-Br):**				
**KI67 > 4%**		43	26.06%	**PWB**		26 (22–28)		
**Side (Hemisphere)**				**SWB**		20,57 (18–20,57)		
**Left**		88	53.33%	**EWB**		23 (19–24)		
**Right**		71	43.03%	**FWB**		22 (18–27)		
**Midline/Bilateral**		6	3.64%	**BrCS**		81.65 (75.27–85.79)		
**Surgical parameters**				**Overall quality of life (QoL)**		169.57 (157.93–183.07)		
**Skull base location**		61	36.97%	**Biological/Functional assessment**				
**Infratentorial location**		20	12.12%	**ASA**	**1 or 2**		131	79.39%
**Max diameter**	1.86 (0.57–2.94)				**3 or 4**		34	20.61%
**Diameter > 25 mm**		54	32.73%	**Pre-operative KPS**		90 (80–90)		
**Preoperative tumour volume (mL)**	27.93 (8.54–44.10)							
**Frailty index (FI)**	0.16 (0.06–0.18)							
**FI profiles**				**Fit**			69	41.82%
				**Semi-Fit**			77	46.67%
				**Frail**			19	11.52%

(Left) The demographic, anatomical, biological, and clinical characteristics of the population are reported in the table. ASA and FI scores were categorised in groups according to the previous literature. (Right) Post-operative and follow-up records are reported in the table. Complications occurrence and its clarification is made by means of the Clavien–Dindo grading scale as reported (0: no complications; 1: Any deviation from the normal post-operative course without the need for pharmacological or surgical treatments; 2: complications requiring pharmacological treatment with drugs or blood/platelet transfusions; 3: complications requiring surgical, endoscopic or radiological intervention; 4: life-threatening complication requiring IC/ICU-management; 5: deceased). Post-operative parameters comprehended prolonged operation time; delayed discharge from ICU and ward; and occurrence of infections, seizures, pulmonary embolisms, or intraparenchymal hemorrhages (ICH). Follow-up parameters comprehended the extent of resection, KPS, quality of life assessment, and mortality at several intervals. ASA: American Society of Anesthesiologists; KPS: Karnofsky performance status; FI: frailty index. PWB: physical well-being; SWB: social well-being; EWB: emotional well-being; FWB: functional well-being; BrCS: brain cancer subscale.

**Table 3 cancers-14-03065-t003:** Multivariable logistic regression analysis.

		**KPS Postop—20 Drop**				**KPS FU—PRE ≤ 20 + QOL < 183**			
		**Univariable**		**Multivariable**		**Univariable**		**Multivariable**	
**Parameters**		**OR (95% C.I.)**	***p* Value **	**OR (95% C.I.)**	***p* Value **	**OR (95% C.I.)**	***p* Value **	**OR (95% C.I.)**	***p* Value **
**Demographics**	**Age > 65**	5.890 (0.673–51.568)	0.109			2.310 (0.994–5.369)	0.002 **	0.420 (0.127–1.392)	0.156
	**Age > 70**	4.235 (0.751–23.880)	0.102			1.960 (0.858–4.481)	0.111		
	**Age > 80**	9.600 (1.778–21.835)	0.009 **	0.720 (0.038–13.569)	0.827	2.841 (0.965–5.363)	0.002 **	0.956 (0.212–4.312)	0.953
	**Previous surgery**	1.390 (0.154–12.515)	0.769			2.218 (0.776–6.339)	0.137		
**Clinical and Functional**	**ASA Score**	22.414 (2.523–39.139)	0.005 **	5.553 (1.760–7.642)	0.023 *	2.616 (1.074–5.368)	0.034 *	2.620 (0.815–8.429)	0.106
	**KPS pre-op**	0.682 (0.539–0.862)	0.001 **	//		0.923 (0.863–0.988)	0.021 *	//	
	**KPS pre-op < 80**	15.800 (11.523–19.498)	<0.001 ***	1.46 (0.017–7.494)	0.98	16.320 (1.631–26.269)	0.017 *	4.824 (0.274–85.050)	0.282
	**Frailty Index (FI)**	4.100 (1.595–10.538)	0.003 **	//		3.107 (1.744–5.534)	<0.001 ***	//	
	**FI > 0.10 (Semi-Fit)**	0.560 (0.100–3.145)	0.510			1.983 (0.865–4.549)	0.016 *	12.479 (2.764–16.349)	0.001 **
	**FI > 0.20 (Frail)**	8.937 (1.663–28.032)	0.011 *	14.752 (1.463–148.777)	0.022 *	4.582 (1.643–8.778)	0.004 **	35.457 (25.210–41.318)	<0.001 ***
**Surgical**	**Skull base location**	9.196 (1.048–20.669)	0.045 *	4.232 (0.280–63.975)	0.050 *	1.607 (0.707–3.654)	0.002 **	0.821 (0.228–2.961)	0.763
	**Infratentorial location**	3.917 (0.669–22.922)	0.013 *	6.079 (1.573–9.282)	0.028 *	4.167 (1.515–9.457)	0.006 **	7.514 (1.514–37.280)	0.014 *
	**Diameter > 2.5 cm**	11.224 (1.277–18.625)	0.029 *	16.078 (0.939–27.310)	0.050 *	2.899 (1.264–6.651)	0.012 *	4.983 (1.720–14.440)	0.003 **
	**Diameter > 3 cm**	3.543 (0.685–18.331)	0.013 *	4.363 × 10^6^ (0.000—//)	0.989	1.764 (0.722–4.308)	0.213		
	**Diameter > 4 cm**	6.950 (1.312–16.828)	0.023 *	2.754 × 10^9^ (0.000—//)	0.998	1.925 (0.683–5.424)	0.215		
**Radiological**	**Calcification**	1.051 (0.186–5.935)	0.955			1.206 (0.512–2.841)	0.668		
	**Severe peritumoral edema**	1.714 (0.335–8.784)	0.518			3.238 (1.394–7.522)	0.006 **	4.162 (1.299–13.331)	0.016 *
	**Necrosis**	2.177e^–8 (0.000—//)	0.995			1.059 (0.419–2.677)	0.904		
	**Hyperostosis**	1.527 (0.269–8.678)	0.633			1.000 (0.389 2.571)	0.997		
	**Heterogeneous Gd enhancement**	2.167 (0.350–3.406)	0.004 **	2.251 (0.348–14.570)	0.394	1.232 (0.508–2.990)	0.04 *	0.850 (0.281–2.567)	0.773
	**Sinus invasion**	1.935 (0.339–11.057)	0.004 **	2.064 (0.313–13.603)	0.451	2.560 (1.046–6.265)	0.039 *	4.458 (1.392–14.279)	0.012 *
	**Tumor shape (Multilobated > 2)**	1.679 (0.298–9.449)	0.557			1.500 (0.638–3.526)	0.353		
	**DWI hyperintensity**	1.829 (0.248–13.470)	0.553			2.303 (0.887–5.981)	0.037 *	3.208 (1.040–9.891)	0.042 *
	**Absence of a Tumor-Brain cleft**	4.567 (0.771–7.056)	0.044 **	5.910 (0.880–39.675)	0.047 *	2.138 (0.687–6.650)	0.001 **	4.350 (1.006–18.818)	0.049 *
				**Nagelkerke R²: 0.560** **AIC: 40.695**				**Nagelkerke R²: 0.347** **AIC: 129.927**	

All of the given parameters were entered into a single variable logistic regression model and were tested for association with the outcomes investigated in the study (post-operative KPS reduction of at least 20 points and follow-up KPS score 20 points lower than baseline plus a quality of life assessment under the 75th percentile of the overall population). Those reporting significance at the univariable regression analysis entered a multivariable logistic regression model to test for independent association. Significant results are followed by an asterisk (*). * *p* ≤ 0.05; ** *p* ≤ 0.01; *** *p* ≤ 0.001. Quality of model fitting is reported by means of the Nagelkerke R^2^ and Akaike information criterion (AIC). OR: odd ratio; CI: confidence interval; QOL: quality of life.

**Table 4 cancers-14-03065-t004:** MBSS Part A and Part B scores: classification performance and sub-group analysis.

**Milan Biometric Surgical Score for Intracranial Meningiomas (MBSS-Men Score; Part A)**								
**ITEM**	MEASURE	SCORE VALUE						
**ASA Score**	1–2	0						
	>2	3						
**Frailty Index**	<0.10	0	**Multivariate regression analysis**		**Odd Ratio (OD)**	**Standard Error (S.E.)**	***p* Value **	**95% C.I.**
	0.10–0.20	2						
	>0.20	3	**Post-operative prognostic Score**	**MBSS (Part A)**	2.611	0.293	0.001	1.469–4.640
**Skull base location**	Yes	1		Constant	0	3.056	0	
	No	0						
**Infratentorial location**	Yes	3	** *AUR-ROC analysis* **		**Area (AUC)**	**Std. Error (S.E.)**	***p* Value **	**95% C.I.**
	No	0						
**Diameter > 2.5 cm**	>25 mm	3		**Overall population**	0.956	0.034	0	0.890–1.022
	<25 mm	0		**Age 18–65 years**	0.878	0.042	0	0.794–0.961
**Tumor-Brain cleft on T2WI**	Absent	2		**Age > 65 years**	0.981	0.017	0	0.948–1.013
	Present	0		**Age > 70 years**	0.973	0.024	0	0.926–1.020
	**RANGE**	0–15		**Age > 80 years**	0.911	0.074	0	0.765–1.057
**Milan Biometric Surgical Score for intracranial meningiomas (MBSS-Men Score; Part B)**								
**Frailty Index**	<0.10	0						
	0.10–0.20	1						
	>0.20	3						
**Infratentorial location**	Yes	2						
	No	0	** *Multivariate regression analysis* **		**Odd Ratio (OD)**	**Standard Error (S.E.)**	***p* Value **	**95% C.I.**
**Diameter > 2.5 cm**	>25 mm	1						
	<25 mm	0	**Follow-up prognostic Score**	**MBSS (Part B)**	2.961	0.203	0	1.988–4.411
**Severe peritumoral edema**	Yes	1		Constant	0.004	0.876	0	
	No	0						
**Sinus Invasion**	Yes	1	** *AUR-ROC analysis* **		**Area (AUC)**	**Std. Error (S.E.)**	***p* Value **	**95% C.I.**
	No	0						
**DWI hyperintensity**	Present	1		**Overall population**	0.877	0.033	0	0.811–0.942
	Absent	0		**Age 18–65 years**	0.901	0.04	0	0.823–0.978
**Tumor-Brain cleft in T2WI**	Absent	1		**Age > 65 years**	0.854	0.054	0	0.749–0.959
	Present	0		**Age > 70 years**	0.85	0.055	0	0.741–0.959
	**RANGE**	0–10		**Age > 80 years**	0.861	0.088	0	0.689–1.033

(Left) The parameters independently associated with the outcomes investigated were selected and filled into a newly designed prognostic score. The Milan Biometric Surgical Score (MBSS) was split in two subparts: Part A comprehended all of the significant predictors of early post-operative functional outcome, while Part B comprehended those predictors affecting long-term functional outcome and quality of life. A chart for MBSS was designed and each parameter was given a value according to its odds ratio on a multivariable regression analysis. MBSS-Part A had a range from 0 to 15. MBSS-Part B had a range from 0 to 10. (Right) The current table reports the classification analysis of MBSS Part A and Part B in predicting early post-operative functional deterioration and long-term unfavourable outcome at follow-up, respectively. Each sub-score was tested first on the overall population and then on the age-specific subgroups.

**Table 5 cancers-14-03065-t005:** Scores comparison.

**Post-Operative Outcome Scores Comparison**				
	**Area (AUC)**	**Std. Error (S.E.)**	***p* Value **	**95% C.I.**
**MBSS (Part A)**	0.956	0.034	0.0001	0.890–1.022
**MCS**	0.724	0.051	0.0001	0.623–0.825
**CRGS ***	0.943	0.030	0.0001	0.885–1.000
**CCI**	0.551	0.096	0.594	0.363–0.740
**FI**	0.752	0.129	0.005	0.500–0.972
**Follow-up Outcome Score comparison**				
	**Area (AUC)**	**Std. Error (S.E.)**	***p* Value **	**95% C.I.**
**MBSS (Part B)**	0.877	0.033	0.0001	0.811–0.942
**MCS**	0.553	0.054	0.328	0.447–0.659
**CRGS ***	0.671	0.053	0.001	0.566–0.775
**CCI**	0.598	0.049	0.046	0.502–0.695
**FI**	0.729	0.054	0.0001	0.623–0.834

The current table reports the classification reports (AUC—area under the curve) at the receiver operating characteristic analysis (AUC-ROC) of MBSS (Part A and B) compared with CRGS, MCS, and CCI. FI is also reported for additional discussion. As reported in the manuscript, FI resulted in being highly predictive of post-operative and follow-up outcomes after resection of the intracranial meningioma. For comparison, the AUC-ROC was computed for the frailty index (FI) score as an independent predictor of post-operative and follow-up outcomes as per se. Previously validated scores (CRGS, MCS, and CCI) reported a moderate performance in predicting the given outcomes. The combination of frailty assessment and anatomical-surgical parameters (MBSS) yielded the highest classification performance, according to our findings. MBSS: Milan biometric surgical score; CCI: Charlson comorbidity index; MCS: Milan complexity scale; CRGS: clinical–radiological grading system. FI: frailty index. * CRGS was designed to be inversely related to an unfavourable outcomes as reported by Caroli et al., and a classification analysis was conducted separately to reproduce an AUC over the reference line (not shown). The latter permitted direct comparison with MBSS, MCS, CCI, and FI.

## Data Availability

The datasets generated and analysed during the current study are available from the corresponding author upon reasonable request.

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
