# Peer review of "A Frailty-Adjusted Stratification Score to Predict Surgical Risk, Post-Operative, Long-Term Functional Outcome, and Quality of Life after Surgery in Intracranial Meningiomas"

_cancers, 2022, doi:10.3390/cancers14133065_

Round 1
Reviewer 1 Report
The manuscript titled “A frailty-adjusted stratification score to predict surgical risk, post-operative, long-term functional outcome and quality of life after surgery in intracranial meningiomas” describes the authors wanted to use a newly designed score to predict intracranial meningiomas patients’ condition before and after the surgery. The followings are some concerns and comments have been pointed out that the authors may want to consider.
- Line 2 Title: The designed score in this manuscript has never been verified or has never been used to predict any other group of the population, or even the population in this study has not been predicted (the score was designed after the authors already knew the results). I think the “predict” is not suitable in the title. However, the authors could discuss, that it might be used to predict other studies in the future.
- Please homogenous the format throughout the manuscript should be seriously revised. For example, with or without a space between the word and the citation serial number; the fonts, and so on.
- Line 32: It seems there is an extra space before “The ASA”. Please double-check throughout the manuscript. For example, line 36 before the word “Conclusion”; line 44 before the word “frailty”, line 73 before the word “the”, and so on.
- Line 38 keywords: None of “skull base surgery” appears in the main context. “Functional assessment” only appears one time in the main context, one time in Table 2. I do not think they are suitable to be keywords. Please consider switching to another one.
- Line 40 Introduction section: this part need to be enriched with more background information.
- Line 57 Materials and Methods section: The authors stated “MBSS was the first score ever designed” line 301. Please provide a detailed method of how to obtain the score to make your work reproducible.
- Line 82 Table 1: To make the 34 items clearer, I would suggest the authors consider adding serial numbers aside.
- Lines 89-91 Figure 1: Please make the figure title and figure legend together. Provide a clearer figure image.
- Line 106: I’d suggest the authors set the equation in an individual line to make it clearer or add “;” to separate it.
- Line 110: The “2” should be superscript in “mm2”. Please double-check.
- Line 149 Results section: I highly recommend the authors generate a figure to present the “outcome” results to make them clearer and more readable.
- Line 151: The “70,3%” should be “70.3%”. Please check throughout the manuscript.
- Line 161 Table 2: a): I’d suggest the authors included total case number 165 in the table, and present each category as “count/165” for easier tracking. b): The population age less than 65 years old is missing in the table. Please double-check. c): Please check the count number for “Pre-operative KPS” should be “165” instead of “175”.
- Line 177 Table 3: a): Please adjust the table to make “***” in the same line. b): Why the population less than 65 years old is excluded?
- Line 201 Table 4: What is the value in the third column stand for? Please update the information in the table to make it clearer.
- Line 216: Please use italic p as it refers to a p-value. And please homogenous the format with or without a space before and after the signs, “=” or “<”. Check throughout the whole manuscript.
- Line 217: The “p <) 0.009” should be “p < 0.009”
- Line 244 Table 5: In the “Follow-up Outcome Score comparison” part, for example, the values in the middle of the table should be “0.553” or “0.671” etc., instead of “.553” or “.671” etc.; check throughout the manuscript.
- Lines 296-297: Where are the Table 6, Table 7, and Table 8?
- There is only one table in the supplementary file, just use “Supplementary Table 1” instead of with an extra title “Supplementary tables”.
Author Response
First, we would like to express our gratitude to the editorial team to have had this manuscript peer-reviewed in Cancers. We really appreciated the possibility to share our findings and discuss the methodology hereby applied and its implications with the reviewers who agreed to further process our study report. Concordantly, we would like to thank the reviewers and their comments. We appreciated your deep understanding on the background we designed our study onto and your careful examination of our manuscript. We believe this peer-review process helped increase the quality of the study itself and the readability of the contents reviewed in the sections of the manuscript.
Hereby follow our comments and modification to the draft according to reviewers’ concerns.
Reviewer 1 #
- Line 2 Title: The designed score in this manuscript has never been verified or has never been used to predict any other group of the population, or even the population in this study has not been predicted (the score was designed after the authors already knew the results). I think the “predict” is not suitable in the title. However, the authors could discuss, that it might be used to predict other studies in the future.
Answer: As correctly underlined by the reviewer the MBSS score has never been used to predict any other population groups in our sample, as the patients affected by meningioma we reviewed were implicated in the design of the model itself. However, the aim of the study was to produce evidences about the importance of included variables in predicting the functional outcome of meningiomas in patients of any age with particular attention to multidomain characteristics as Frailty. We believe our preliminary findings might be of interest to colleagues and readers to test the consistency of the score and eventually apply it in new cohorts of patients. A prospective assessment is ongoing at our Institution and will be the subject of future work.
- Please homogenous the format throughout the manuscript should be seriously revised. For example, with or without a space between the word and the citation serial number; the fonts, and so on.
Answer: All format issues throughout the manuscript have been revised and corrected.
- Line 32: It seems there is an extra space before “The ASA”. Please double-check throughout the manuscript. For example, line 36 before the word “Conclusion”; line 44 before the word “frailty”, line 73 before the word “the”, and so on.
Answer: All format issues throughout the manuscript have been revised and correct.
- Line 38 keywords: None of “skull base surgery” appears in the main context. “Functional assessment” only appears one time in the main context, one time in Table 2. I do not think they are suitable to be keywords. Please consider switching to another one.
Answer: 20 out of 165 patients were classified as infratentorial and 61 as skull base tumours. We found the keyword “skull base surgery” largely suitable to the contents of the manuscript. “Functional assessment” relied on the characteristics of the two outcome measures implicated in the study design as reported in Line 181-190: “The primary post-operative outcome (“Early post-operative functional deterioration”) was designed to address patient dependence status and was computed as a drop in post-operative KPS of at least 20 points at discharge compared to the pre-operative assessment. The selected cutoff well represented the occurrence of any general or neurological complication affecting the overall functional performance of patients undergoing surgery[21,35]. The long-term follow-up primary outcome (“Unfavorable long term functional autonomy and quality of life”) was designed to address patient dependence and QoL and was defined as a decrease of ³ 20 points in KPS at the follow-up interview compared to the pre-operative one plus an overall quality of life (QOL) under the 75th percentile of the examined population. “
- Line 40 Introduction section: this part need to be enriched with more background information.
Answer: Introduction section was extended and a clearer overview of the background topic and recent literature findings is now available [Line 62-96].
- Line 57 Materials and Methods section: The authors stated “MBSS was the first score ever designed” line 301. Please provide a detailed method of how to obtain the score to make your work reproducible
Answer: References were added to the Method section to enhance reproducibility of the score and better define variables of interest. In addition, a specific checklist was produced to carried out all the variables included in MBSS Part A and Part B. The checklist is reported in Supplementary Table 2.
- Line 82 Table 1: To make the 34 items clearer, I would suggest the authors consider adding serial numbers aside.
Answer: Serial numbers aside were added in Table 1 to increase readability of the 34 items FI checklist.
- Lines 89-91 Figure 1: Please make the figure title and figure legend together. Provide a clearer figure image.
Answer: Figure title was included in the Figure. An higher resolution image was provided.
- Line 106: I’d suggest the authors set the equation in an individual line to make it clearer or add “;” to separate it.
Answer: All format issues throughout the manuscript have been revised and corrected.
- Line 110: The “2” should be superscript in “mm2”. Please double-check.
Answer: All format issues throughout the manuscript have been revised and corrected.
- Line 149 Results section: I highly recommend the authors generate a figure to present the “outcome” results to make them clearer and more readable.
Answer: Figures order and contents were reviewed. A Figure 1 was produced: it comprehends an Histogram describing the distribution of MBSS subscores among those patients experiencing or not an early post-operative or long-term functional deterioration and reduced quality of life. The histogram on the left represents the median MBSS Part A scores among those experiencing a decrease of at least 20 points in post-operative KPS or not compared to the pre-operative assessment.. The histogram on the right represents the median MBSS Part B scores among those experiencing a decrease of at least 20 points in follow-up KPS plus an overall quality of life measurement (FACT-Br overall score) under the 75th percentile of the population under exam or not compared to the pre-operative assessment. 95% C.I. Error bar are visualised.
- Line 151: The “70,3%” should be “70.3%”. Please check throughout the manuscript.
Answer: All format issues throughout the manuscript have been revised and corrected.
- Line 161 Table 2: a): I’d suggest the authors included total case number 165 in the table, and present each category as “count/165” for easier tracking. b): The population age less than 65 years old is missing in the table. Please double-check. c): Please check the count number for “Pre-operative KPS” should be “165” instead of “175”.
Answer: Total case number is now reported in Table 2. As the topic of the study focuses on elderly patients, we further reviewed our data exploring the effect of the variables of interest in the elderly subpopulation. However, an overview of the overall sample characteristics is now available and age classed were split as 18-64, 65-80 and older than 80. Pre-operative KPS total assessments count of “175” was corrected with “165”
- Line 177 Table 3: a): Please adjust the table to make “***” in the same line. b): Why the population less than 65 years old is excluded?
Answer: As previously reported, the focus of the study was to assess the predictability of certain variables of interest in elderly and frail patients. Our population included a proportion of younger patients but the regression analyses was carried out to assess the predictive effect of aging (Age.> 65 at least) on the functional outcomes and quality of life as reported in the manuscript. The weighted importance of these variables on younger patients might be the topic of further investigations.
- Line 216: Please use italic p as it refers to a p-value. And please homogenous the format with or without a space before and after the signs, “=” or “<”. Check throughout the whole manuscript.
Answer: All format issues throughout the manuscript have been revised and corrected.
- Line 217: The “p <) 0.009” should be “p < 0.009”
Answer: All format issues throughout the manuscript have been revised and corrected.
- Line 244 Table 5: In the “Follow-up Outcome Score comparison” part, for example, the values in the middle of the table should be “0.553” or “0.671” etc., instead of “.553” or “.671” etc.; check throughout the manuscript
Answer: All format issues throughout the manuscript have been revised and corrected.
- Lines 296-297: Where are the Table 6, Table 7, and Table 8?
Answer: The typos were corrected and enumeration of tables is now coherent with the contents reported in the manuscript.
- There is only one table in the supplementary file, just use “Supplementary Table 1” instead of with an extra title “Supplementary tables”.
Answer: The Supplementary File now contains Supplementary Tables 1 and 2. Legends are reported below each of the tables to enhance readability.
Reviewer 2 Report
Dear Authors, I read with pleasure this interesting manuscript entitled "A frailty-adjusted stratification score to predict surgical risk, post-operative, long-term functional outcome and quality of life after surgery in intracranial meningiomas".
The aim of the study is clear and notable, authors conducted a detailed analysis of all possible factors affecting early and long-time outcomes in patiens with meningiomas surgically trated.
Authors included several factors from wich they assumed a predictive score MILAN BIO(METRIC)-SURGICAL SCORE (MBSS).
The manuscript is properly written with detailed descriptions of all methodological steps. There are, however, some concerns.
The score appears too complex to be routinely applied; the main aim of a score is being quick to be completed and to be figured out, both for physicians and patients.
Authors should have tested their score prospectively also on a still cohort of patients to validate it. Moroever, all patiens coming from a single center and there could be a selection bias ( i.e assessing FI, ASA score, perilesional edema etc.)
Authors should test the score also in patients who were excluded from surgery due to high FI or ASA score, to better test their finding.
The cohort of patients included is too small to draw any conlcusions. However, the aim of the study is really interesting: evaluate the patients from different point of view, especially for the elderly ones, in a period where age is not considered as a downside anymore.
Minor typos should be addressed
Author Response
First, we would like to express our gratitude to the editorial team to have had this manuscript peer-reviewed in Cancers. We really appreciated the possibility to share our findings and discuss the methodology hereby applied and its implications with the reviewers who agreed to further process our study report. Concordantly, we would like to thank the reviewers and their comments. We appreciated your deep understanding on the background we designed our study onto and your careful examination of our manuscript. We believe this peer-review process helped increase the quality of the study itself and the readability of the contents reviewed in the sections of the manuscript.
Hereby follow our comments and modification to the draft according to reviewers’ concerns.
Reviewer 2 #
We thank Reviewer 2 for his/her/* comments and judgment to our manuscript. We reviewed some of the aspects highlighted reviewing the comments provided by Reviewer 1 but we will go through them again to answer all reasonable concerns and implications pointed out. The MBSS was designed according to the multivariable regression analysis computed on the population included in our study. The score was designed to predict two different outcomes (early post-operative and long-term checkpoints). Taken separately, MBSS Part A and Part B do not appear of extreme complexity and their scoring is supposed to imply less of 5 minutes to be computed once all necessary information have been gathered. Also, Part A and Part B – according to their relevance – are designed to be computed in two different checkpoints of post-operative care (before discharge and during long term follow-up interviews). The instructions summarized in Supplementary Table 2 should help MBSS look more reproducible in clinical practice.
The selection bias and limitations given the retrospective and monocentric nature of the study have been carefully discussed in the manuscript. Further prospective validations are necessary to hypothesize a clinical use of MBSS score. However, to the best of our knowledge, inter-personal assessment bias of FI have rarely been reported in literature. Nevertheless, the monocentric quality of the investigation might define a limitation in the generalizability of the score.
Testing a new prognostic score in patients not suitable to undergo surgery or those who refuse surgical interventions has ever been subject of debate in previous published studies. However, in clinical practice, some of the information deemed necessary to compute prognostic scores under validation are not available in those patients who were not eligible to surgery and followed up in outpatient clinic settings. Furthermore, asymptomatic meningiomas might be numerous in the latter group of patients. As widely accepted, these cases have a different trajectory of outcome as they do not require aggressive treatments in most of the cases over time. Given so, comparation between patients undergoing surgery and those who did not require such an intervention might be of limited use in our cases but should be pursued in population studies to assess the real quality of life and independence status in the “non-surgical” subclass of patients.
Finally, our limited population size limited the relevance of our findings. However, we believe that the publication of our findings might provide preliminary but convincing evidences about the relevance of several biological, anatomical and radiological features in predicting functional outcomes in elderly patients undergoing surgery for intracranial meningiomas.
To conclude, we again thank the editorial staff and the reviewers for this opportunity to review our manuscript and discuss the major findings and concerns.
Should any additional questionable issue pointed out, we would be obliged to review it to positively affect the quality of our study.
Round 2
Reviewer 1 Report
I do not have any further concerns now, except for the following minors, and please homogenous the format throughout the manuscript again before publication. Good luck.
1. Line 60: There is an extra “.”.
2. Line 62: The number value “12.500” should be “12,500”.
3. Line 199: The “1,8%” should be “1.8%”.
4. Line 241: Please italic p.
Author Response
We thank again Reviewer #1 for his/her/* revision of our manuscript.
I uploaded the final version of the study as requested.
All formatting issues in the manuscript have been reviewed and solved.
Thank you for your peer-review and the chance to optimise our study report.
Sincerely,
The authors.